# Biochemical properties of naturally occurring human bloom helicase variants

**Rachel R. Cueny** [1], **Sameer Varma** [2,3], **Kristina H. Schmidt** [2]*, **James L. Keck** [1]*

**1** Department of Biomolecular Chemistry, University of Wisconsin, Madison, WI, United States of America, **2** Department of Cell Biology, Microbiology, and Molecular Biology, University of South Florida, Tampa, FL, United States of America, **3** Department of Physics, University of South Florida, Tampa, FL, United States of America

* kschmidt@usf.edu (KHS); jlkeck@wisc.edu (JLK)

**Data Availability Statement:** Raw data for all experiments and replicates are available to all readers via Dryad using doi:10.5061/dryad.5tb2rbp7t.

## Abstract

Bloom syndrome helicase (BLM) is a RecQ-family helicase implicated in a variety of cellular processes, including DNA replication, DNA repair, and telomere maintenance. Mutations in human *BLM* cause Bloom syndrome (BS), an autosomal recessive disorder that leads to myriad negative health impacts including a predisposition to cancer. BS-causing mutations in *BLM* often negatively impact BLM ATPase and helicase activity. While *BLM* mutations that cause BS have been well characterized both *in vitro* and *in vivo*, there are other less studied *BLM* mutations that exist in the human population that do not lead to BS. Two of these non-BS mutations, encoding BLM P868L and BLM G1120R, when homozygous, increase sister chromatid exchanges in human cells. To characterize these naturally occurring BLM mutant proteins *in vitro*, we purified the BLM catalytic core (BLM$_{core}$, residues 636–1298) with either the P868L or G1120R substitution. We also purified a BLM$_{core}$ K869A K870A mutant protein, which alters a lysine-rich loop proximal to the P868 residue. We found that BLM$_{core}$ P868L and G1120R proteins were both able to hydrolyze ATP, bind diverse DNA substrates, and unwind G-quadruplex and duplex DNA structures. Molecular dynamics simulations suggest that the P868L substitution weakens the DNA interaction with the winged-helix domain of BLM and alters the orientation of one lobe of the ATPase domain. Because BLM$_{core}$ P868L and G1120R retain helicase function *in vitro*, it is likely that the increased genome instability is caused by specific impacts of the mutant proteins *in vivo*. Interestingly, we found that BLM$_{core}$ K869A K870A has diminished ATPase activity, weakened binding to duplex DNA structures, and less robust helicase activity compared to wild-type BLM$_{core}$. Thus, the lysine-rich loop may have an important role in ATPase activity and specific binding and DNA unwinding functions in BLM.

## Introduction

Bloom syndrome (BS) is an autosomal recessive disorder that is characterized by shorter than average stature, sensitivity to sunlight, weakened immune response, and a predisposition to cancer [1]. Patients with BS are more likely to encounter age-associated complications early in

**Funding:** This work was supported an award from the US National Institutes of Health/National Institute of General Medical Sciences (R01 GM139296) to KHS and JLK. The funders had no role in study design, data collection and analysis, decision to publish, or preparation of the manuscript.

**Competing interests:** The authors have declared that no competing interests exist.

life, including adult-onset diabetes, various cancers, and chronic obstructive lung disease [1–3]. BS can be diagnosed by assessing the level of damage and instability in chromosomes. BS is characterized by elevated chromosomal breakage, increased sister chromatid exchanges (SCEs), and quadriradial chromosomal structures [4]. Previous research has found that BS-causing mutations lead to a 10–12 fold increase in SCEs in human lymphocytes, and similar phenotypes have been demonstrated in DT40 lymphoma cells [5, 6].

BS is caused by mutations to *BLM*, which encodes the Bloom Syndrome DNA helicase (BLM) [4]. BLM is a member of the RecQ family of helicases and can unwind a variety of DNA secondary structures with 3′ to 5′ polarity, including double-stranded DNA (dsDNA), forked DNA substrates, G-quadruplex DNA (G4s), and Holliday junctions (HJs) [3, 7]. There are five human RecQ-family helicases, which include BLM, WRN, RECQL1, RECQL4, and RECQL5. Mutations in four of these RecQ helicases, including *BLM*, are associated with autosomal recessive disorders [8, 9]. As such, RecQ-family helicases are often defined as "caretakers" of the genome [10].

BLM is essential for promoting genome stability in the cell and has functions in DNA replication, DNA repair, transcription, and telomere maintenance [3]. In keeping with its role as a genome maintenance protein, BLM localizes to DNA replication and DNA damage sites, and to telomeres [3]. Further, BLM interacts with a large network of DNA replication and repair proteins and is thought to act in concert with a larger complex of proteins for many of its core functions [3, 11]. For example, BLM is part of the dissolvasome, which is composed of BLM, topoisomerase IIIα (Top3α), RecQ-mediated genome instability protein 1 (RMI1), and RMI2 [11]. The dissolvasome is involved in dissolution of double Holliday junctions (dHJs) generated during homologous recombination. This function of the dissolvasome prevents crossover during recombination, suppressing SCEs [12, 13]. Thus, defects in BLM cause many adverse effects on chromosome stability due to the multifaceted nature of BLM.

The BLM catalytic core includes a Helicase domain, a RecQ C-terminal domain (RQC), and a Helicase and RNase D C-terminal domain (HRDC) (Fig 1A) [14, 15]. The Helicase domain of BLM is composed of two RecA-like subdomains, which are important for ATP hydrolysis and helicase activity. The RQC domain is composed of Zinc-binding (ZN) and Winged Helix (WH) subdomains. The ZN subdomain includes four cysteine residues that coordinate a $Zn^{2+}$ ion, and the WH is important for interactions with various DNA structures. The HRDC is likely involved in binding single-stranded DNA (ssDNA) and increasing BLM processivity [7]. The N-terminal and C-terminal regions of BLM are highly disordered and expected to be important for protein-protein interactions [7, 16].

Mutations in *BLM* that lead to BS include missense mutations, nonsense mutations, frameshifts, and splicing defects [4]. Many of the missense mutations interfere with helicase and ATPase activity of BLM [4, 17]. While the impact of homozygous mutations in *BLM* have been extensively characterized clinically, the health impacts of heterozygous BS-causing mutations is less well-studied. Limited studies have found that heterozygous BS mutations can lead to increased cancer risk in humans and in mice [18, 19]. Thus, although BS is recessive and manifests only when both copies of *BLM* are mutated, disrupting activity of one *BLM* allele may still have detrimental impacts on human health and longevity.

In addition to the lack of study on the impacts of heterozygous *BLM* mutations, there has been very little clinical or *in vitro* study of other naturally occurring *BLM* mutations that do not result in BS. There are over 1400 *BLM* mutations that exist in the human population, with 62 encoding for pathogenic mutations, 118 mutations considered benign, and the rest of uncertain or conflicting human health relevance [20]. In 2012, Mirzaei and Schmidt [21] investigated the effects of 41 *BLM* single nucleotide polymorphisms from the Short Genetic Variations database map by assessing the effects of these mutations in *Saccharomyces*

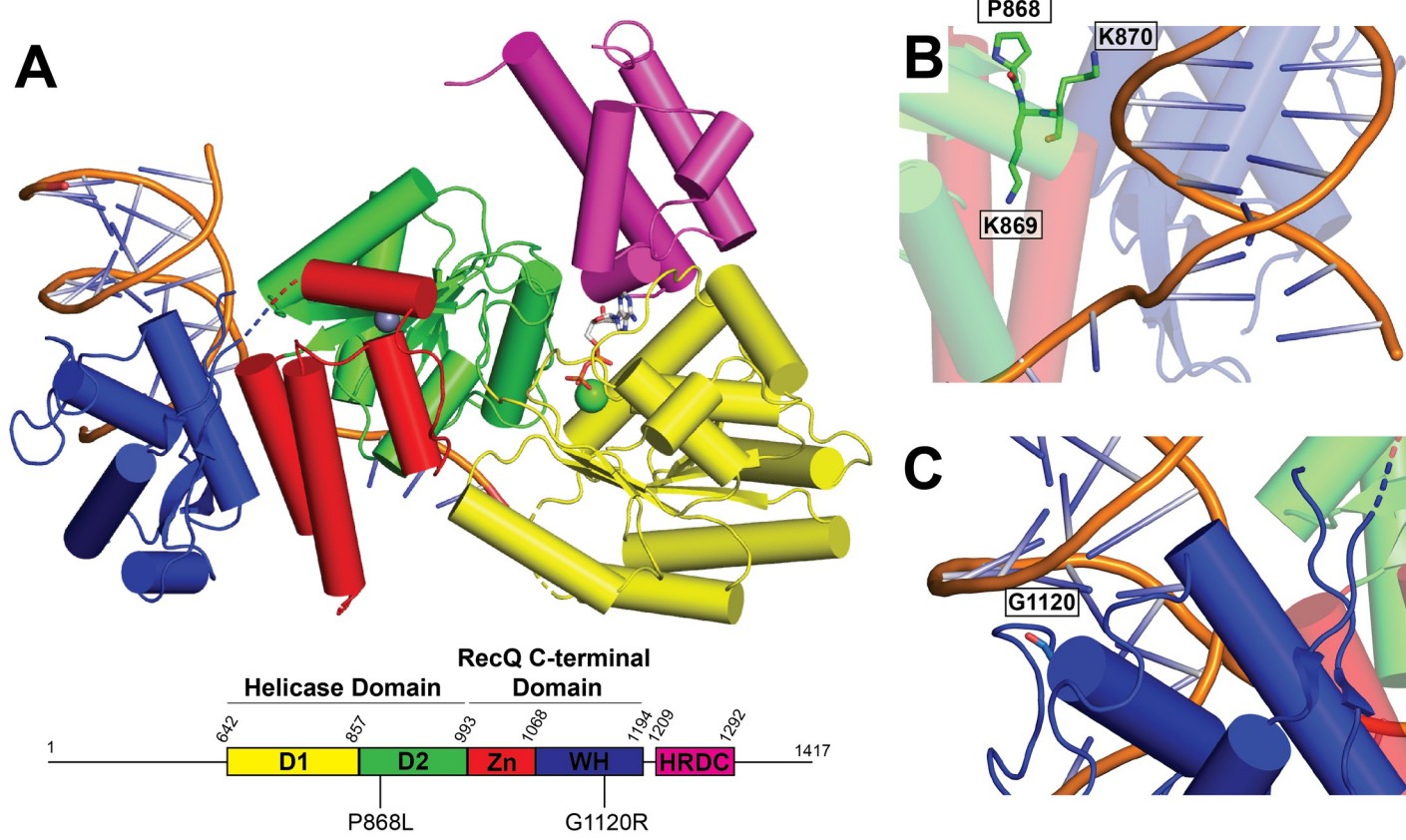

**Fig 1. Structure of BLM.** (A). Structure of BLM catalytic core (PDBID: 4O3M [14]), colored. according to domain map below structure. ADP is shown as sticks and magnesium and zinc ions are shown as green and grays spheres, respectively. Positions of substitutions P868L and G1120R are highlighted on domain map. (B). P868, K869, and K870 residues, shown as sticks. (C). G1120 residue, shown as sticks.

*cerevisiae.* Interestingly, two substitutions, P868L and G1120R, led to a partial loss in function, demonstrated by increased sensitivity to hydroxyurea compared to wild-type (WT) BLM. These mutations also cause a 4–5 fold increase in SCEs, a moderate increase in sensitivity to hydroxyurea at low concentrations in human cells, and a decreased clearance of phosphory-lated histone H2AX after treatment with camptothecin in human cells [22]. While these *BLM* mutations have less severe phenotypes than BS-causing mutations, P868L and G1120R substitutions still increase cellular genomic instability. Mutations in *BLM* that cause this intermediate phenotype could lead to increased risk of cancer or other adverse health impacts in humans. Interestingly, the *BLM* P868L allele has a 5.13% frequency in the human population, and the impacts of this heterozygous (or homozygous) *BLM* allele on human health and longevity are unknown [22]. Residue P868 resides in the helicase domain of BLM (Fig 1B) and is part of a lysine-rich loop (868-PKKPKK) that appears to be important for interacting with DNA at the ssDNA-dsDNA junction in BLM crystal structures [14, 15]. Residue G1120 is part of the BLM WH and terminates an alpha helix preceding a loop proximal to dsDNA in the BLM-dsDNA structure (Fig 1C).

To better understand the effects of naturally occurring non-BS *BLM* mutations, we tested BLM$_{core}$ constructs (residues 636–1298) with P868L or G1120R substitutions using a battery of biochemical assays. We also measured activity of a BLM$_{core}$ K869A K870A mutant protein to assess the role of lysines within the BLM lysine-rich loop. The BLM$_{core}$ retains helicase, DNA

binding, and ATPase activity, making it the simplest model for characterizing the impacts of these naturally occurring mutant proteins *in vitro* [14, 15]. Previous *in vitro* studies investigating the impacts of BS-causing mutations utilized a nearly identical BLM catalytic core construct (residues 642–1290) and found that BS-causing mutations had severe impacts on the helicase and ATPase activity of BLM [17]. Additionally, defects detected with BLM P868L and BLM G1120R in yeast used a construct that included residues 648–1417 of BLM [21]. Therefore, we focused our experiments on BLM$_{core}$ constructs to allow comparison with prior reports and to allow measurement of possible moderate impacts of the mutations on *in vitro* function of BLM. In addition to our biochemical experiments, we compared molecular dynamics (MD) simulations of the WT BLM$_{core}$ and BLM$_{core}$ P868L bound to DNA and ADP.

While we predicted that the P868L and G1120R substitutions would impact the *in vitro* function of BLM$_{core}$, these mutant proteins had similar DNA-dependent ATPase activity, DNA binding activity, and helicase activity to WT BLM$_{core}$. In contrast, BLM$_{core}$ K869A K870A requires ~4-fold more DNA to stimulate ATPase activity and has less robust helicase activity compared to WT BLM$_{core}$. Additionally, DNA binding experiments suggest that BLM$_{core}$ K869A K870A may be deficient in recognizing ssDNA-dsDNA junctions. Thermal stability measurements indicate that differences in the activity of BLM$_{core}$ K869A K870A could not be explained by general destabilization associated with the mutations. Since the naturally occurring BLM$_{core}$ mutant proteins maintain activity levels that are very similar to WT BLM$_{core}$, our results indicate that these mutations lead to moderate genome instability through impairment of other cellular functions of BLM. In alignment with these results, the MD simulations suggest that the P868L substitution could subtly weaken interaction of DNA with BLM's winged-helix domain and alter the orientation of the N-terminal lobe of the ATPase domain, which could shed light on previously identified defects of the hypomorphic P868L mutant protein *in vivo* [22].

## Materials and methods

### Protein expression and purification

The BLM$_{core}$ (residues 636–1298) and BLM$_{core}$ mutant proteins were overexpressed as previously described [14]. Briefly, overexpression was carried out in Rosetta 2 (DE3) *E. coli* transformed with pLysS and BLM$_{core}$ or BLM$_{core}$ mutant protein overexpression plasmids. Cells were grown at 37˚C in Terrific Broth supplemented with 50 μg/mL kanamycin and 50 μg/mL chloramphenicol until cultures reached an $OD_{600}$ ~1.8. BLM overexpression was induced using 0.5 mM IPTG and cells were grown overnight at 18˚C. Cells were then pelleted and stored at -80˚C.

Cells pellets were resuspended in lysis buffer (20 mM Tris-HCl, pH 8.0, 0.5 M NaCl, 10% (v/v) glycerol, 0.1% Triton X-100, 20 mM imidazole, 1 Pierce Protease inhibitor tablet/100 mL buffer (Thermo Fisher), and 0.1 mM phenylmethylsulfonyl fluoride), lysed using sonication, and cleared via centrifugation. Clarified lysate was loaded onto a 5 mL HisTrap FF column (Cytiva) that was equilibrated with Buffer A (20 mM Tris-HCl, pH 8.0, 0.5 M NaCl, 10% (v/v) glycerol, and 20 mM imidazole). Protein was eluted with Buffer B (20 mM Tris-HCl, pH 8.0, 0.5 M NaCl, 10% (v/v) glycerol, 250 mM imidazole). Fractions containing BLM$_{core}$ were diluted to 0.25 M NaCl and loaded onto a HiPrep 16/10 Heparin FF Column (Cytiva). The column was washed with Buffer C (20 mM Tris-HCl, pH 8.0, 0.25 M NaCl, 10% (v/v) glycerol) and then BLM$_{core}$ was eluted from the column using Buffer D (20 mM Tris-HCl, pH 8.0, 1 M NaCl, 10% (v/v) glycerol). Fractions containing BLM$_{core}$ were concentrated in Vivaspin-20 10 kDa concentrator (Sartorius) to 2 mL and loaded onto HiPrep 16/60 Sephacryl S-300 HR column (Cytiva) using Sizing Buffer (20 mM Tris-HCl, pH 8.0, 0.5 M NaCl, 5% (v/v) glycerol).

Fractions containing $BLM_{core}$ were concentrated to ~10 mg/mL then flash frozen in liquid nitrogen and stored at -80˚C. $BLM_{core}$ and mutant proteins were tested for nuclease activity and found to have no detectable nuclease activity (S1 Fig).

The $BLM_{core}$ K869A K870A construct was purified with the following modifications to remove a contaminating nuclease. After collecting fractions from the HisTrap FF column, fractions were assessed for nuclease activity. Five μL of each fraction was incubated for 30 minutes at ambient temperature with 40 nM fluorescein labeled $dT_{30}$ in 50 mM Tris-HCl, pH 7.5, 50 mM KCl, 1 mM DTT, 0.1 mg/mL Bovine Serum Albumin (BSA), and 5 mM $MgCl_2$. Five μL of stop buffer (2% SDS, 5 μg/mL proteinase K, 20% (v/v) glycerol, 0.1 ethylenediamine-tetraacetic acid (EDTA)) was added to each sample, and 5 μL of each sample was loaded onto a 15% acrylamide 1.5-mm gel in Tris-Borate-EDTA (TBE) buffer supplemented with 100 mM KCl. Gels were run at 75 V for 1 hour at 4˚C in 1x TBE running buffer with 100 mM KCl and imaged on an Azure c600 (Azure Biosystems). Fractions that contained significant nuclease activity were discarded, and fractions containing $BLM_{core}$ K869A K870A without significant nuclease activity were diluted to 0.15 M NaCl and loaded onto the HiPrep 16/10 Heparin FF Column, washed with Buffer C2 (20 mM Tris-HCl, pH 8.0, 0.15 M NaCl, 10% (v/v) glycerol), and eluted over a 20-column volume gradient up to 0.5 M NaCl. Fractions containing $BLM_{core}$ were tested for nuclease activity. Fractions that contained $BLM_{core}$ and no detectable nuclease activity were concentrated in Vivaspin-20 10 kDa concentrator (Sartorius) to 2 mL and loaded onto the HiPrep 16/60 Sephacryl S-300 HR column (Cytiva) using Sizing Buffer (20 mM Tris-HCl, pH 8.0, 0.5 M NaCl, 5% (v/v) glycerol). Fractions were tested for nuclease activity, and fractions with no detectable nuclease activity were concentrated then flash frozen in liquid nitrogen and stored at -80˚C.

## Differential scanning fluorimetry

Differential scanning fluorimetry (DSF) was carried out as previously described with the following modifications [23]. WT $BLM_{core}$, $BLM_{core}$ P868L, $BLM_{core}$ G1120R, or $BLM_{core}$ K869A K870A (5 μM) were incubated in DSF buffer (50 mM Tris-HCl, pH 7.5, 50 mM KCl, 1 mM DTT, and 5 mM $MgCl_2$) with 5X Sypro Orange (Millipore-Sigma) in 20 μL total volume for 10 minutes at room temperature. For reactions containing ADP or ATPγS, nucleotide was added to final concentration of 0.5 mM. Samples were then heated from 4˚C to 95˚C at a rate of 0.25˚C $s^{-1}$ with fluorescence measured every two seconds using the HEX filter in a C1000 Touch Thermal Cycler (BioRad). Each DSF trial was performed in triplicate.

DSF data were analyzed using DSFWorld (https://gestwickilab.shinyapps.io/dsfworld/) [24]. Raw fluorescence data were trimmed from 30˚C to 60˚C and the maximum of the first derivative (dRFU) of the denaturation curve is reported as the melting temperature ($T_m$). $\Delta T_m$ values for ADP and ATPγS samples were determined by subtracting the average $T_m$ for each protein alone from the $T_m$ value measured for the same protein in the presence of ADP or ATPγS. Significance was determined using Welch's two-tailed t-test using the default settings on Prism Version 9.3.1. Normalized fluorescence plots were produced using data trimmed from 30˚C to 60˚C in Prism Version 9.3.1.

## DNA-dependent ATPase assay

WT $BLM_{core}$, $BLM_{core}$ P868L, or $BLM_{core}$ G1120R (5 nM) were incubated with either no DNA or $dT_{20}$ serially diluted from 10 nM to 9.8 x $10^{-3}$ nM in ATPase buffer (20 mM Tris-HCl, pH 8.0, 50 mM NaCl, 5% (v/v) glycerol, 0.1 mM DTT, 5 mM $MgCl_2$, 0.1 mg/mL BSA, 2 mM 2-phosphoenolpyruvate, 3 U/mL Pyruvate Kinase/Lactate Dehydrogenase, 0.2 mM NADH). $BLM_{core}$ K869A K870A at 5 nM was incubated with no DNA or with $dT_{20}$ serially diluted from

**Table 1. Oligonucleotides used in this study.**

| | |
|---|---|
| FAM-oRC96 | `5'-GGGTTAGGGTTAGGGTTAGGGTTTTTTTTTT-FAM-3'` |
| oRC32 | `5'-TGGCGACGGCAGCGAGGCTTAGGGTTAGCGTTAGCG TTAGGGTTTTTTTTTTTTTTT-3'` |
| oRC75 | `5'-TGGCGACGGCAGCGAGGCTTAGGGTTAGGGTTAGGG TTAGGGTTTTTTTTTTTTTTTTTT-3'` |
| oAV320 | `5'-GCCTCGCTGCCGTCGCCA-FAM-3'` |
| oAV322 | `5'-GCCTCGCTGCCGTCGCCA-3'` |

100 nM to 0.98 nM in ATPase buffer. Reactions were initiated by addition of 1 mM ATP and $A_{340 nm}$ was monitored for 1 hour at 25˚C. Data were analyzed as previously described [25] and plotted using Prism Version 9.3.1. ATPase assays were done in triplicate. Significance was determined using Welch's two-tailed t-test using the default settings on Prism Version 9.3.1. For standard error of the mean (SEM) measurements, these values were calculated in Prism Version 9.3.1 by dividing the standard deviation by the square root of the number of replicates.

## Electrophoresis mobility shift assays

Electrophoresis mobility shift assays (EMSAs) were performed as previous described [26] with the following modifications. Partial duplex DNA (oRC32/FAM-oAV320, Table 1) or G4 DNA (FAM-oRC96, Table 1) constructs were folded by incubating 5 μM DNA in 10 mM Tris-HCl, pH 7.5 and 100 mM KCl at 95˚C for 10 minutes and slowly cooling the sample to room temperature over several hours. DNA constructs were stored at 4˚C. Serial dilutions of $BLM_{core}$ or $BLM_{core}$ mutant proteins were incubated with 40 nM partial duplex DNA, G4 DNA, or FAM-$dT_{30}$ in 50 mM Tris-HCl, pH 7.5, 50 mM KCl, 1 mM DTT, 0.1 mg/mL BSA, and 5 mM $MgCl_2$ for 30 minutes at room temperature. 3.3% (v/v) glycerol was added to samples, and 5 μL of each sample was loaded onto a 5% acrylamide 1.5-mm gel in TBE buffer supplemented with 100 mM KCl. Gels were pre-run at 75 V for 20 minutes before loading protein/DNA complexes and running at 75 V for 1 hour at 4˚C in 1xTBE running buffer supplemented with 100 mM KCl. Gels were imaged on the Azure c600 (Azure Biosystems). Experiments were done in triplicate.

## Helicase assay

Helicase assays were performed as previous described [26] with the following modifications. Partial dsDNA or G4-dsDNA constructs were folded by incubating 5 μM DNA in 10 mM Tris-HCl, pH 7.5 and 100 mM KCl at 95˚C for 10 minutes and slowly cooling the sample to room temperature over several hours, then stored at 4˚C. Serial dilutions of $BLM_{core}$ or $BLM_{core}$ mutant proteins were incubated with 40 nM oRC32/FAM-oAV320 (partial duplex, Table 1) or oRC75/FAM-oAV320 (G4-dsDNA, Table 1) in helicase assay buffer (50 mM Tris-HCl, pH 7.5, 50 mM KCl, 1 mM DTT, 0.1 mg/mL BSA, 5 mM $MgCl_2$, 5 mM ATP, 4 μM oAV322) for 15 minutes at 37˚C in 25 μL reactions. Folded DNA control was obtained by incubating reaction mixture without $BLM_{core}$, and melted control was obtained by omitting $BLM_{core}$ and heating reaction to 95˚C for 10 minutes. Five μL of stop buffer (2% SDS, 5 μg/mL proteinase K, 20% (v/v) glycerol, 0.1 mM EDTA) was added to each reaction and 5 μL of each sample was loaded onto a 15% acrylamide 1.5-mm gel in TBE buffer supplemented with 100 mM KCl. Gels were run at 75 V for 1 hour at 4˚C in 1x TBE running buffer with 100 mM KCl. Gels were imaged on an Azure c600 (Azure Biosystems). $BLM_{core}$ dsDNA, $BLM_{core}$ P868L dsDNA and G4-dsDNA, and $BLM_{core}$ K869A K870A dsDNA and G4-dsDNA were done in triplicate. $BLM_{core}$ G4-dsDNA, $BLM_{core}$ G1120R dsDNA and G4-dsDNA were done in

quadruplicate. One data point was removed from the BLM$_{core}$ G1120R G4-dsDNA dataset (at BLM$_{core}$ G1120R concentration of 1.95 nM) after visual inspection of the gel which had additional fluorescence on the gel that obscured the results of the lane. DNA unfolding was quantified using ImageJ and analyzed using Prism Version 9.3.1 and fitting data to Eq (1). Significance was determined using Welch's two-tailed t-test using the default settings on Prism Version 9.3.1.

$$Fraction\ unwound = Fraction\ unwound_{max} * \frac{[BLM]}{(K_{BLM} + [BLM])} + Background \qquad (1)$$

## Molecular dynamics simulations

The initial conformation for WT BLM$_{core}$ was taken from one of the X-ray structures (PDB ID: 4CGZ) [15]. This structure contains a portion of the BLM protein (residues 637–1290) bound to DNA, an ADP molecule and a Zn$^{2+}$ ion. Coordinates of the missing hydrogens were generated using the 'pdb2gmx' module of the GROMACS software [27], and the initial coordinates of a missing loop (residues 1194–1206) were constructed using MODELLER [28]. All titratable amino acids were assigned their default protonation states at pH of 7.4, except for the four cysteines (residues 1036,1055,1063,1066) that coordinate the bound Zn$^{2+}$ ion, which are expected to be deprotonated [29]. Histidine protonation assignments (HD1 or HE2) were selected by carrying out a global optimization of the hydrogen bond network [30]. After building all missing atoms, the complex was energy minimized and then placed in a 12×12×12 nm cubic box containing 54,282 water molecules, 97 K$^+$ ions and 79 Cl$^-$ ions, corresponding to an ionic strength of 100 mM. The difference in the numbers of K$^+$ and Cl$^-$ ions serves to counter balance the net negative change of -18 eu on the complex. Following addition of solvent, the system was energy minimized for 500 steps, after which it was subjected to molecular dynamics (MD) simulation. The initial conformation of the P868L mutant protein was constructed from the energy minimized structure of the WT complex. BLM P868L was placed in a unit cell identical to the WT complex and it contains the same numbers of water molecules and salt ions. The P868L mutant protein unit cell was also energy minimized for 500 steps, and then subjected to MD.

All energy minimization and MD simulations were carried out using GROMACS version 2020 [27]. MD integration was carried out using the leap-frog algorithm and with a time step of 2 fs. All bonds were constrained [31]. MD simulations were carried out under isothermal-isobaric conditions. Pressure was regulated at 1.01325 bar using the Parrinello-Rahman extended ensemble approach [32], coupling time constant of 1.0 ps, and a compressibility of $4.5 \times 10^{-5}$. Temperature was regulated at 310 K using the velocity-rescale thermostat [33] and a coupling constant of 0.1 ps, which uses a stochastic term for the generation of a proper canonical ensemble. Periodic boundary conditions were set in all directions, and electrostatic interactions were computed using a particle mesh Ewald scheme [34] with a short-range cutoff of 10 Å, Fourier grid spacing of 1 Å, and a fourth-order interpolation. van der Waals interactions were computed explicitly for interatomic distances smaller than 10 Å. Neighbor lists were constructed using grid search and were updated every 5 steps. Less frequent updates to neighbor lists, which are typically implemented to accelerate simulations on GPUs, resulted in DNA deformation. Water is described using the SPC/E model [35], and ion-water interactions are described using Joung and Cheatham parameters [36]. Protein, solvent ions, and DNA are described using the AMBER99SB-ILDN force field [37], along with NB-fix corrections for protein-DNA, ion-DNA and ion-ion interactions [38]. Note that the Joung and Cheatham ion parameters are not parameterized to reproduce reference data for ion-protein interactions [39], however, their use is justified as ions are not expected to bind directly to protein.

## Results

### BLM$_{core}$ K869A K870A requires higher levels of ssDNA than WT BLM$_{core}$ to stimulate ATPase activity

BLM requires ATP hydrolysis for translocation along DNA and helicase activity. Therefore, the human BLM mutant proteins BLM P868L and BLM G1120R could cause cellular defects due to a decrease in ATPase activity. To test this, we measured the DNA-dependent ATPase activity of mutant BLM$_{core}$ proteins. We used a dT$_{20}$ ssDNA substrate for this assay to examine only ssDNA-dependent activity, avoiding complications that could arise from ATPase activity associated with dsDNA unwinding. Interestingly, both of the BLM$_{core}$ mutant proteins had similar maximum ATPase rates to WT BLM$_{core}$ (Fig 2A, Table 2), indicating that BLM$_{core}$ P868L and BLM$_{core}$ G1120R are able to efficiently hydrolyze ATP when bound to ssDNA. While BLM$_{core}$ P868L had an apparent increase in maximum ATPase rate compared to WT BLM (P value = 0.0116), there is a relatively small difference (35%) between the WT BLM$_{core}$ and BLM$_{core}$ P868L maximum ATPase rates (Table 2). Since the concentration of BLM$_{core}$ directly impacts measured maximal ATPase rates, small inaccuracies in concentration determination can lead to modest apparent differences that are difficult to interpret. We therefore do not ascribe biochemical significance to this difference in maximum ATPase rate. The DNA concentration-dependence of the ATPase activity was also similar for WT BLM$_{core}$, BLM$_{core}$ P868L, and BLM$_{core}$ G1120R. The DNA concentrations required for half-maximum ATPase rate (K$_{DNA}$) differed by less than 1-fold among all three proteins (Fig 2B, Table 2).

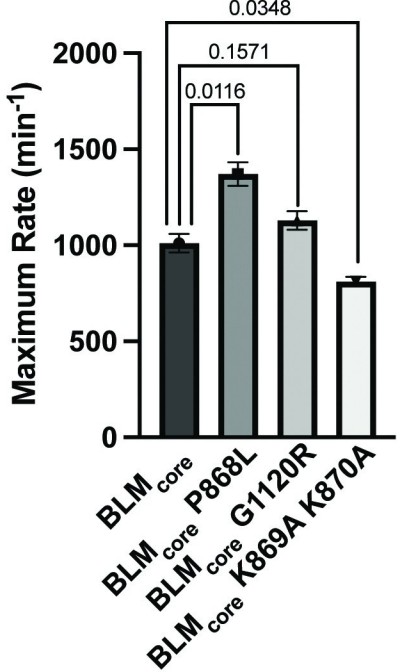

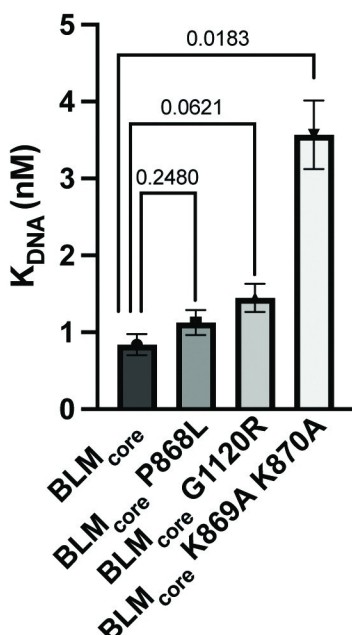

**Fig 2. DNA-dependent ATPase assays.** (A). Maximum DNA-dependent ATPase rates for WT BLM$_{core}$, BLM$_{core}$ P868L, BLM$_{core}$ G1120R, and BLM$_{core}$ K869A K870A. These data represent the mean of three replicates with error bars representing the standard error of the mean. P-values are indicated above the bars. (B). ssDNA concentrations required for half-maximum ATPase rate (K$_{DNA}$) for WT BLM$_{core}$, BLM$_{core}$ P868L, BLM$_{core}$ G1120R, and BLM$_{core}$ K869A K870A. These data represent the mean of three replicates with error bars representing the standard error of the mean. P-values are indicated above the bars.

**Table 2. Summary of ATPase, DNA binding, and helicase activity.**

| ATPase Data Summary | | | | |
|---|---|---|---|---|
| | WT BLM$_{core}$ | BLM$_{core}$ P868L | BLM$_{core}$ G1120R | BLM$_{core}$ K869A K870A |
| Maximum ATPase Rate (min$^{-1}$) | 1010 ± 48.2 | 1370 ± 62.0 | 1130 ± 48.0 | 811 ± 25.0 |
| K$_{DNA}$(nM) | 0.839 ± 0.137 | 1.13 ± 0.163 | 1.45 ± 0.185 | 3.57 ± 0.444 |
| ATPase rate at 0 nM DNA (min$^{-1}$) | 110 ± 5.4 | 78 ± 17 | 45 ± 3.7 | Not detected |
| **DNA Binding Data Summary** | | | | |
| | WT BLM$_{core}$ | BLM$_{core}$ P868L | BLM$_{core}$ G1120R | BLM$_{core}$ K869A K870A |
| dsDNA with 3′ ssDNA overhang | 79 nM, distinct | 62.5 nM, distinct | 62.5 nM, distinct | 62.5 nM, smear |
| G4 with 3′ ssDNA overhang | 125 nM, smear | 125 nM, smear | 250 nM, smear | 625 nM, smear |
| ssDNA | 125 nM, smear | 62.5 nM, smear | 62.5 nM, smear | 312.5 nM, smear |
| **Helicase Activity Summary** | | | | |
| | WT BLM$_{core}$ | BLM$_{core}$ P868L | BLM$_{core}$ G1120R | BLM$_{core}$ K869A K870A |
| K$_{BLM}$ for dsDNA substrate unwinding (nM) | 15 ± 2.4 | 12 ± 2.4 | 13 ± 2.4 | 22 ± 6.6 |
| Maximum fraction of dsDNA unwound | 0.99 ± 0.038 | 1.1 ± 0.058 | 0.92 ± 0.043 | 0.76 ± 0.060 |
| K$_{BLM}$ for G4-dsDNA substrate unwinding (nM) | 17 ± 2.4 | 6.2 ± 1.4 | 23 ± 3.1 | 46 ± 8.8 |
| Maximum fraction of G4-dsDNA unwound | 0.96 ± 0.031 | 0.95 ± 0.051 | 0.99 ± 0.031 | 0.72 ± 0.032 |

Summary of ATPase, DNA binding, and helicase assay data described in this study. For DNA binding data, values indicated are the estimated apparent K$_D$ which either manifested as a smear or distinct band shift. ATPase and helicase measurements include standard error of the mean values calculated using Prism Version 9.3.1.

BLM$_{core}$ K869A K870A was found to have a modestly different maximum ATPase rate compared to WT BLM$_{core}$ (P value = 0.0348). More interestingly, this mutant protein had a ~4-fold higher K$_{DNA}$ value (P value = 0.0183), indicating that higher concentrations of ssDNA were necessary to stimulate its half-maximum ATPase activity (Fig 2, Table 2). Unlike maximal ATPase measurements, K$_{DNA}$ values are independent of protein concentration, indicating that this difference is biochemically significant. These data show that disrupting the charge of the lysine-rich loop impacted the concentration of ssDNA required to stimulate DNA-dependent ATP hydrolysis in BLM$_{core}$ whereas the P868L and G1120R substitutions did not. This result is consistent with a reduced DNA binding affinity for BLM$_{core}$ K869A K870A relative to the other BLM$_{core}$ constructs tested.

We also assessed the ATPase rate for each mutant protein in the absence of DNA. WT BLM$_{core}$, BLM$_{core}$ P868L, and BLM$_{core}$ G1120R were capable of hydrolyzing ATP in the absence of DNA, with rates of 4–10% of that observed with saturating levels of ssDNA. In contrast, BLM$_{core}$ K869A K870A had no detectable DNA-independent ATPase activity (Table 2). This finding aligns with the higher DNA concentration requirement for this mutant protein and may point to a defect in its overall ATPase functions.

## BLM$_{core}$ K869A K870A has reduced specificity for binding ssDNA-dsDNA junctions

BLM is able to bind and unwind a variety of DNA structures [3]. To test the ability of the BLM$_{core}$ proteins to bind to different DNA structures, we used Electrophoretic Mobility Shift Assays (EMSAs) to assess binding to ssDNA, G4 DNA with a 3' ssDNA overhang, and dsDNA with a 3' ssDNA overhang (partial dsDNA), each done in triplicate. WT BLM$_{core}$ bound to ssDNA and the G4 substrate, as evidenced by slower migration of the labeled DNA on a gel (Fig 3). This shift manifested as "smearing" for these substrates, which could indicate that these complexes are dynamic, dissociating and re-associating in the assay. In contrast, when tested with partial dsDNA, WT BLM$_{core}$ was able to form a distinct band shift, consistent with

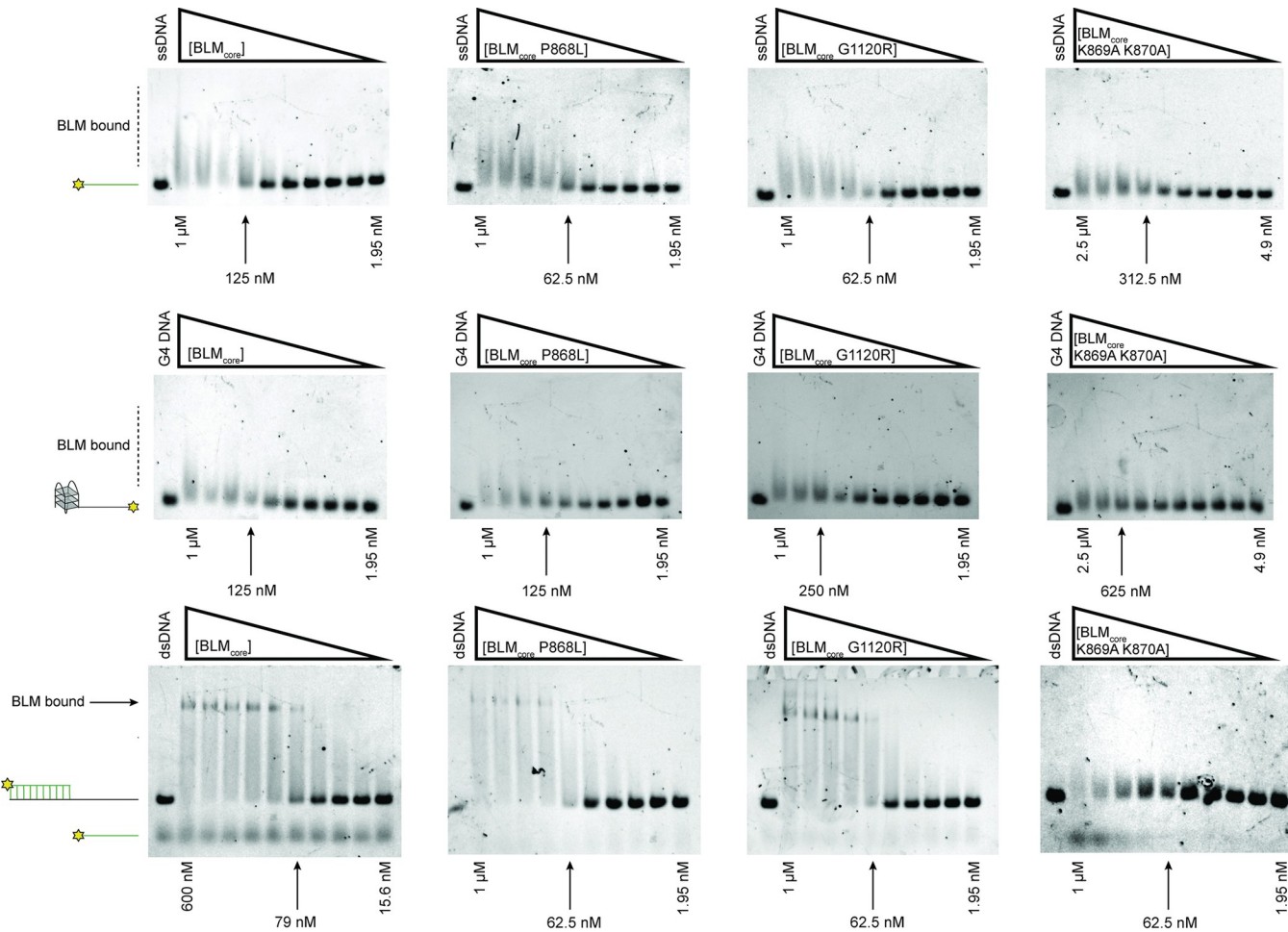

**Fig 3. DNA binding assays for WT BLM and mutant proteins.** Electrophoretic mobility shift assays for WT BLM$_{core}$, BLM$_{core}$ P868L, BLM$_{core}$ G1120R, and BLM$_{core}$ K869A K870A (left to right) for binding ssDNA (top), G4 DNA (middle), and partial-duplex DNA (bottom). Lowest and highest concentrations are indicated below each gel, and the estimated apparent K$_D$ is indicated with an arrow. Concentrations for WT BLM$_{core}$ with G4 and with ssDNA, BLM$_{core}$ P868L with G4, ssDNA, and partial-dsDNA, and BLM$_{core}$ G1120R with G4, ssDNA, and partial-dsDNA are 1.00 μM, 500 nM, 250 nM, 125 nM, 62.5 nM, 31.2 nM, 15.6 nM, 7.81 nM, 3.91 nM, and 1.95 nM. WT BLM with partial-dsDNA concentrations are 600 nM, 400 nM, 267 nM, 178 nM, 118 nM, 79.0 nM, 52.7 nM, 35.1 nM, 23.4 nM, and 15.6 nM. Concentrations for BLM$_{core}$ K869A K870A with G4, ssDNA, and partial-dsDNA are 2.5 μM, 1.25 μM, 625 nM, 312 nM, 156 nM, 78.1 nM, 39.1 nM, 19.5 nM, 9.77 nM, and 4.88 nM. DNA concentration for each assay is 40 nM. EMSAs were run in triplicate with the gel shown here as a representative example.

WT BLM$_{core}$ forming a stable complex with the substrate. The same trend was observed for BLM$_{core}$ P868L and BLM$_{core}$ G1120R, which were both able to shift ssDNA and G4 DNA, indicated by smearing, and were able to shift partial dsDNA, indicated by distinct band shifts (Fig 3). From estimating qualitative binding affinities from the gel shift assays, WT BLM$_{core}$, BLM$_{core}$ P868L, and BLM$_{core}$ G1120R all appear to have similar affinities for each of these DNA substrates (Table 2).

BLM$_{core}$ K869A K870A was also able to bind to ssDNA and G4 DNA (Fig 3) but required higher protein concentrations to shift the DNA, consistent with a lower affinity for these substrates (Table 2). This binding also resulted in a smear above the substrate migration site. Whereas WT BLM$_{core}$, BLM$_{core}$ P868L, and BLM$_{core}$ G1120R were able to bind the partial dsDNA causing a distinct band shift, the BLM$_{core}$ K869A K870A mutant protein did not have this effect. Instead, BLM$_{core}$ K869A K870A binding to partial duplex DNA resulted in a smear,

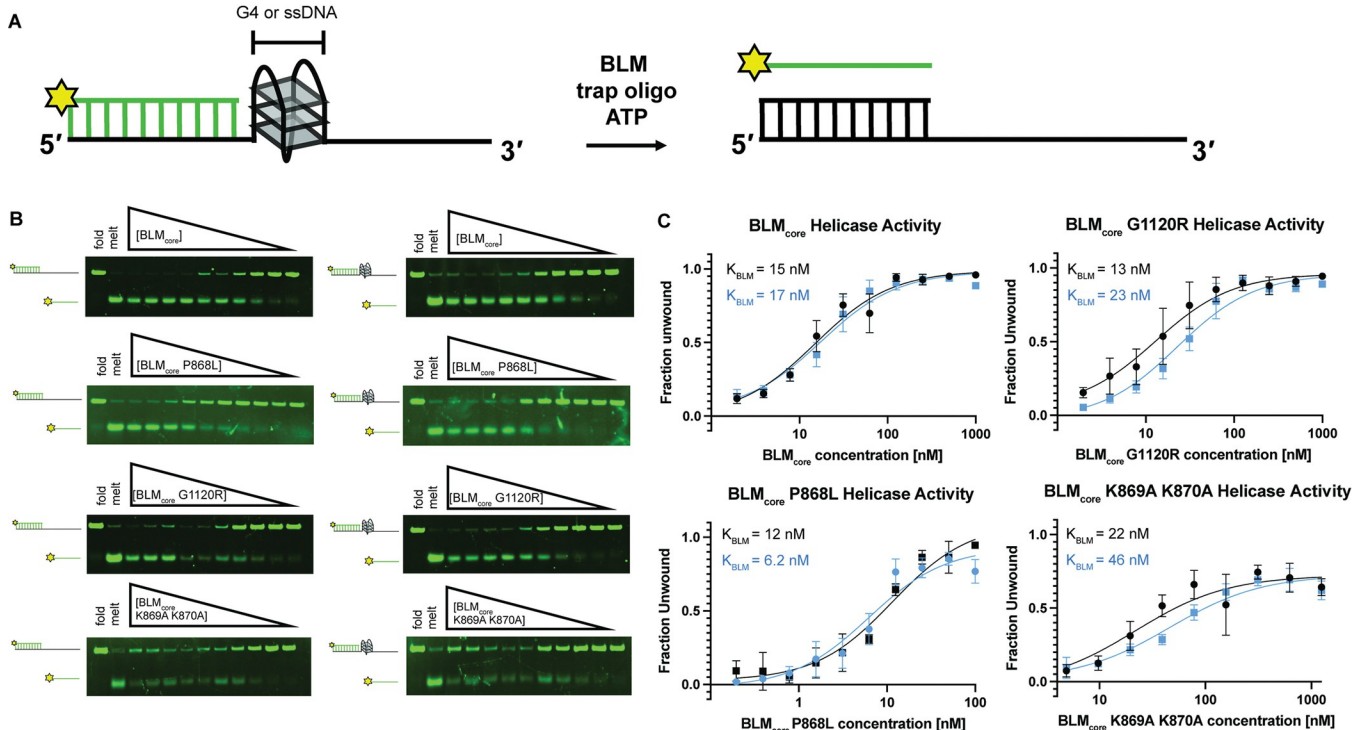

**Fig 4. WT BLM_core and BLM_core mutant protein helicase assays unwinding duplex DNA and G4 DNA.** (A) Schematic of experimental system. (B). (Top to bottom): WT BLM_core, BLM_core P868L, BLM_core G1120R, and BLM_core K869A K870A gel-based helicase assays assessing duplex DNA unwinding (left) and G4-dsDNA unwinding (right). For WT BLM_core and BLM_core G1120R, protein concentrations are 1.00 μM, 500 nM, 250 nM, 125 nM, 62.5 nM, 31.2 nM, 15.6 nM, 7.81 nM, 3.91 nM, and 1.95 nM. For BLM_core P868L, protein concentrations are 100 nM, 50 nM, 25 nM, 12.5 nM, 6.25 nM, 3.12 nM, 1.56 nM, 0.781 nM, 0.391 nM, and 0.195 nM. For BLM_core K869A K870A, protein concentrations are 2.5 μM, 1.25 μM, 625 nM, 312 nM, 156 nM, 78.1 nM, 39.1 nM, 19.5 nM, 9.77 nM, and 4.88 nM. DNA concentrations are 40 nM for each assay. Gels are representative examples of helicase assays run in triplicate or quadruplicate. (C). Quantification of fraction of substrate unwound for dsDNA unwinding (black) and G4-dsDNA unwinding (blue). Data points are the mean values of helicase assays with error bars representing the standard error of the mean.

similar to what was observed for binding to ssDNA and G4 DNA. Additionally, a band corresponding to the labeled ssDNA from the substrate was consistently observed with the highest concentration of BLM_core K869A K870A (1 μM). This could be due to modest ATP-independent unwinding at high concentrations of this mutant protein. As described in the Materials and Methods, this effect is not due to a contaminating nuclease in the BLM_core K869A K870A purification. Thus, BLM_core K869A K870A appears to have reduced overall DNA binding stability and reduced ability to discriminate among different DNA structures.

## BLM_core K869A K870A is a less efficient helicase than WT BLM_core, BLM_core P868L, and BLM_core G1120R

We next tested the ability of the BLM mutant proteins to unwind dsDNA and G4-dsDNA. This experiment was done using a gel-based helicase assay with a substrate containing a 3' ssDNA 15 nucleotide overhang, either the human telomere G4-forming sequence or a control sequence that cannot form a G4, and 18 base pairs of dsDNA (Fig 4A). The annealed strand contains a fluorescent label on its 3' end and unwinding of this substrate can be observed by tracking migration of fluorescent label in a gel (Fig 4B). The reaction is initiated by the addition of BLM_core or mutant protein, ATP, and an unlabeled trap oligo that has the same sequence as the fluorescently labeled ssDNA. The total fraction of DNA unwound at various BLM_core concentrations is measured (Fig 4C), allowing a determination of the fraction of

DNA that $BLM_{core}$ or mutant protein can unwind and the concentration of $BLM_{core}$ required for half-maximum unwinding ($K_{BLM}$).

WT $BLM_{core}$ was able to efficiently unwind both the dsDNA and the G4-dsDNA substrates (Fig 4B and 4C), with over 90% maximum unwinding for both. The $K_{BLM}$ value for WT BLM was 15 ±

2.4 nM and 17 ± 2.4 nM for dsDNA and G4-dsDNA, respectively (Table 2). Both $BLM_{core}$ G1120R and $BLM_{core}$ P868L were able to unwind over 90% of the dsDNA substrate and G4-dsDNA substrate as well, indicating that these mutant proteins have similar maximum activity levels to WT $BLM_{core}$. $BLM_{core}$ P868L had $K_{BLM}$ values of 12 ± 2.4 nM and 6.2 ±1.4 nM for dsDNA and G4-dsDNA, respectively (Table 2), indicating that $BLM_{core}$ P868L requires similar amounts of protein to unwind dsDNA as WT $BLM_{core}$ but that it was somewhat more efficient at unwinding G4-dsDNA substrates (P values = 0.36 and 0.012). $BLM_{core}$ G1120R had $K_{BLM}$ values of 13 ± 2.4 nM and 23 ± 3.1 nM for dsDNA and G4-dsDNA unwinding, indicating that $BLM_{core}$ G1120R unwound dsDNA and G4-dsDNA with a similar efficiency to that of WT $BLM_{core}$ (P values = 0.56 and 0.22).

$BLM_{core}$ K869A K870A was unable to unwind the same percentage of substrate, with lower than 80% total unwinding for both dsDNA and G4-dsDNA (Table 2). At high concentrations of $BLM_{core}$ K869A K870A, we observed inhibition of unwinding (Fig 4C), which is also observed at high concentrations of WT $BLM_{core}$. The $K_{BLM}$ values obtained from $BLM_{core}$ K869A K870A unwinding dsDNA and G4-dsDNA were 22 ± 6.6 nM and 46 ± 8.8 nM, respectively, however standard error of the mean (SEM) was much higher for this mutant protein (Table 2). These values were determined to not be statistically significant when compared to the WT $BLM_{core}$ dsDNA and G4-dsDNA $K_{BLM}$ values (P values = 0.42 and 0.072), but this is likely due to the large SEM for $BLM_{core}$ K869A K870A.

## All $BLM_{core}$ mutant proteins are stably folded under assay temperatures

One possible explanation for the impact of the substitutions in the $BLM_{core}$ K869A K870A protein was that the substitutions destabilize the protein. To test whether sequence changes in the $BLM_{core}$ mutant proteins altered proper protein folding, the melting temperature ($T_m$) of each $BLM_{core}$ mutant protein was measured using differential scanning fluorimetry (DSF). WT $BLM_{core}$ and $BLM_{core}$ P868L have similar $T_m$ values (46.3 ± 0.12°C and 45.7 ± 0.23°C, respectively), whereas $BLM_{core}$ G1120R and $BLM_{core}$ K869A K870A have $T_m$ values of 42.6 ± 0.23°C and 42.1 ± 0.30°C, respectively (S2 Fig, Fig 5A). The $T_m$ differences among the proteins are modest, and all of the proteins are stably folded at temperatures used for all the ATPase assays, DNA binding assays, and helicase assays of this study.

We also tested the impact of ADP and ATPγS on the $T_m$ of $BLM_{core}$ and $BLM_{core}$ mutant proteins. All proteins were stabilized by the presence of ADP or ATPγS (Fig 5B, 5C), indicating that each $BLM_{core}$ protein could bind these nucleotides. Interestingly, $BLM_{core}$ G1120R is stabilized to a greater extent by ADP, with a $\Delta T_{m, ADP}$ of 3.7°C compared to the $\Delta T_{m, ADP}$ of WT $BLM_{core}$ of 2.9°C. $BLM_{core}$ K869A K870A was also stabilized to a greater extent by ADP, with a $\Delta T_{m, ADP}$ of 4.8°C. $BLM_{core}$ K869A K870A was also stabilized to a greater extent by ATPγS, with a $\Delta T_{m, ATPγS}$ of 3.6°C compared to the WT $BLM_{core}$ $\Delta T_{m, ATPγS}$ of 2.1°C. Overall, these data shows that all $BLM_{core}$ mutant proteins are stabilized by nucleotide binding, with $BLM_{core}$ K869A K870A having the greatest increase in $\Delta T_m$ for both ADP and ATPγS.

## Molecular dynamics simulations of $blm_{core}$ P868L

The *BLM P868L* allele has a 5.13% frequency in the human population and causes an increased frequency of sister chromatid exchanges in cell culture, making it essential to better

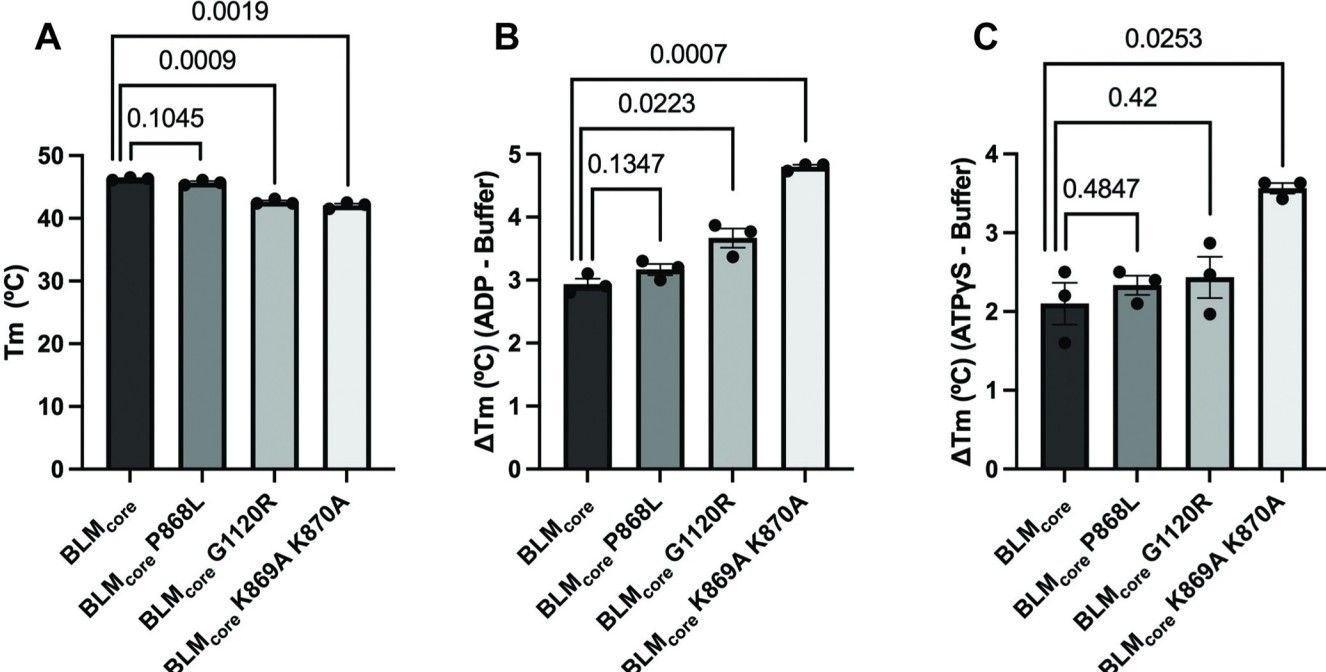

**Fig 5. BLM$_{core}$ mutant protein melting temperatures.** (A) Melting temperature (T$_m$) of BLM$_{core}$, BLM$_{core}$ P868L, BLM$_{core}$ G1120R, and BLM$_{core}$ K869A K870A. Error bars indicate standard error of the mean and p-values are indicated above bars. (B) Change in T$_m$ ($\Delta$T$_{m, ADP}$) for each protein with the addition of 0.5 mM ADP compared to the T$_m$ of each mutant protein without nucleotide present. $\Delta$T$_{m, ADP}$ values were obtained by subtracting the mean T$_m$ in (A) of each mutant protein from each T$_m$ obtained in the presence of ADP. Error bars indicate the standard error of the mean and p-values are indicated above bars. (C) The $\Delta$T$_{m, ATP\gamma S}$ for each protein with the addition of 0.5 mM ATP$\gamma$S compared to the T$_m$ of each mutant protein without nucleotide present. $\Delta$T$_{m, ATP\gamma S}$ values were obtained as in (B). Error bars indicate the standard error of the mean and p-values are indicated above bars.

understand how this substitution impacts BLM function. To examine effects of the P868L substitution on the structure and dynamics of the BLM$_{core}$ complexed with partial dsDNA and ADP, we carried out MD simulations. In MD simulations, macromolecules are placed in bulk solvent and all atoms belonging to macromolecules and solvent are set free to move in three-dimensional space under the influence of each other's forces and under applied boundary conditions of pressure (density) and temperature. During simulations, the structure of the macromolecule is free to change, and the solvent is free to move around. These simulations are run for a fixed period of time, which provides atomic-level insights into the structural evolution and dynamics of macromolecules. In this work, we carried out six 1-μs long MD simulations, three of the BLM$_{core}$ P868L mutant protein and three of the WT BLM$_{core}$ (control). Each of these simulations required about two months of wall clock time on a single-GPU, 20-core workstation.

First, we analyzed the simulation data to examine the association of DNA with BLM. In the X-ray structure of WT BLM$_{core}$ (PDB ID: 4CGZ [15]), the dsDNA (12 bp) interacts directly with the RecA-like lobe 2 of the ATPase domain (RecA-D2) and with the ZN and WH subdomains of the RQC domain of BLM. The single-stranded portion of the DNA (5 nucleotides) interacts directly only with RecA-D2. We find that this general pattern of association between BLM and DNA is maintained in the MD simulations of the WT BLM$_{core}$. The one difference that we note between the X-ray structure and MD simulation is that the distance between DNA and ZN domain is larger in MD simulations (Fig 6A). This is unlikely to be an artifact of the employed force field because we note such increased DNA-ZN distances even when the complex is simulated with an older version of the DNA-protein force field (amber-99sb-ILDN without DNA-protein NB-fix corrections [37]) in which protein-DNA binding is stronger.

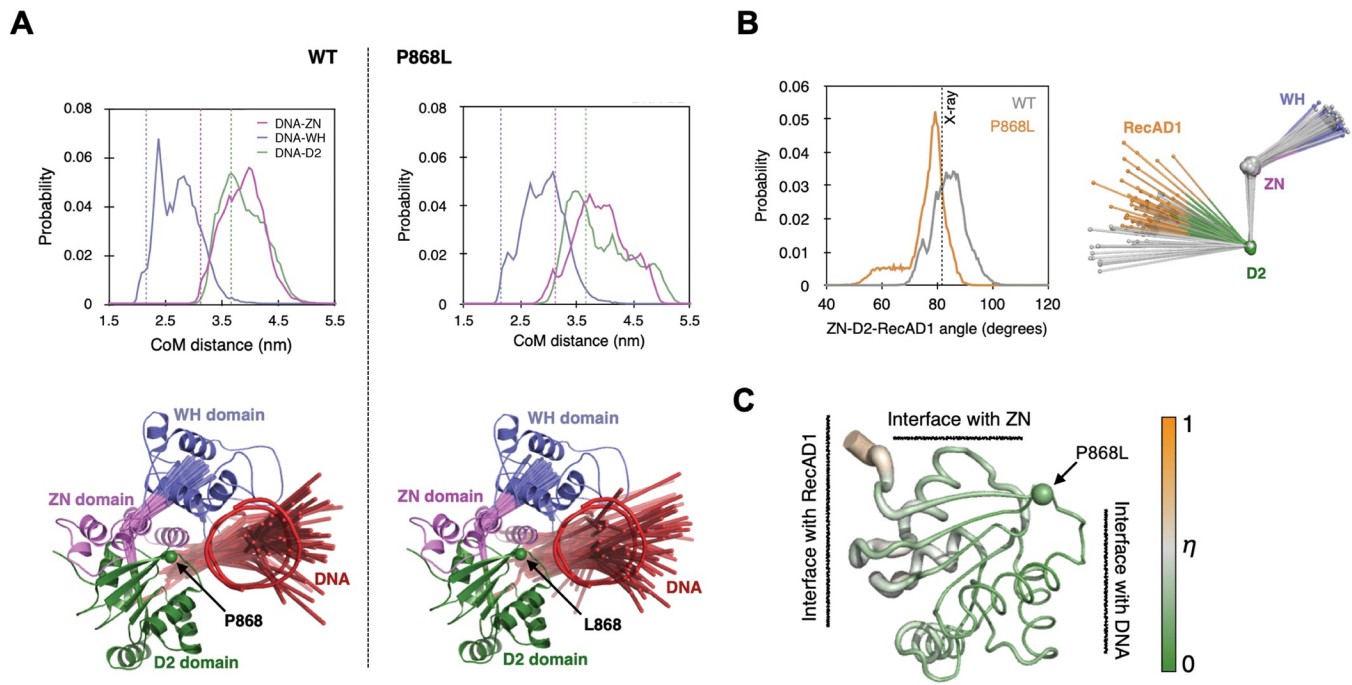

**Fig 6. Effects of the P868L substitution on the structure and dynamics of the BLM$_{core}$ protein complexed with DNA and ADP.** (A) Effect of the P868L substitution on the structure and dynamics of bound DNA. Top panels: Fluctuations in distances between centers of mass. (CoMs) of DNA and BLM domains (ZN-zinc-binding domain, WH-winged helix domain, D2- RecA-like lobe 2 of the ATPase domain). The dashed vertical lines indicate CoM distances in the X-ray structure. Bottom panels: MD snapshots superimposed over theBLM$_{core}$ X-ray crystal structure. The purpose of choosing a coarse-grained representation to illustrate the finding is that in MD, the DNA moves away from the BLM protein. The D2, ZN and WH domains are shown using their CoMs connected by solid lines: the CoM of D2 is connected to the CoM of ZN and the CoM of ZN is connected to the CoM of WH. DNA is shown using two line segments, one connecting the CoMs of the two halves of the double stranded region, and the other connecting the CoM of its single stranded region with the CoM of one half of its double stranded region. Sixty snapshots are shown, twenty taken from each of the three MD simulations at regular time intervals of 50 ns. The coordinates of the CoMs and coarse-grained representation of the DNA are determined after the backbone atoms of D2 in that snapshot are fitted onto the backbone atoms D2 in the X-ray structure. Consequently, the CoM of D2 appears stationary in time. (B) Effect of P868L substitution on the angle between the CoMs of ZN, D2, and D1 domains. The dashed vertical line on the figure on the left is the angle observed in the X-ray structure. The figure on the right compares the distribution of the domain CoMs (spheres) in 60 equally spaced snapshots extracted from MD simulations. The grey spheres are CoMs of domains in the WT BLM and those in other colors are CoMs of domains in the P868L mutant protein. (C) P868L induced shift ($\eta$) in conformational ensemble of the D2 domain. The X-ray structure of the D2 domain is shown as cartoon putty where the width of the putty as well as color of the putty denote the magnitude of $\eta$. Higher values of $\eta$ imply larger effects of P868L induced shifts in conformational ensembles.

Comparison of WT and P868L simulations reveals one key effect of the P868L substitution on DNA-BLM binding: while DNA-ZN and DNA-D2 distances are similar to those in the simulations of the WT form, the DNA-WH distance is larger (Fig 6A). Therefore, effectively, the P868L mutation pushes the DNA further away from the BLM core. This is also visually discernible in the structural coarse-grained representations of BLM-DNA complex in the bottom panel of Fig 6A compared to WT as there are distinctly more conformations of DNA that are further away from BLM in P868L. From an energetics standpoint, this implies that the P868L substitution weakens the interaction between the BLM core and DNA. We note that our current in vitro biochemical experiments do not show any discernible effect of P868L on BLM activity, which appears to be in contrast with both MD simulations and our previous cellular assays (21, 22). This may perhaps be due to the low sensitivity and qualitative nature of the biochemical assays employed in this study.

Other than BLM-DNA interactions, the P868L substitution also changes the orientation of the RecA-D1 relative to the rest of BLM in MD simulations. The P868L substitution reduces the angle between the centers of mass (CoMs) of the D1, D2 and ZN domains (Fig 6B). Since the sites for ATP-binding and hydrolysis are at the interface between RecA-D1 and RecA-D2,

this result suggests potential effects of P868L on ATP hydrolysis and, thus, BLM activity. Changes were not observed in our biochemical assays, but this may be due to use of ssDNA instead of partial dsDNA. However, interpretation of ATPase rate data with partial duplex DNA in the biochemical assay is not simple, since ATPase activity would be coupled to DNA unwinding that would change the structure of the stimulating DNA from partial duplex to ssDNA during steady-state measurements.

To understand how P868L substitution alters RecA-D1 orientation, we examined the effect of P868L substitution on the conformational ensemble of RecA-D2. We used Direct Comparison of Ensembles (DiCE) [40–42], which computes the physical overlap between two conformational ensembles in high-dimensional space. It yields the difference in ensembles in terms of a quantity '$\eta$' that is normalized to vary between 0 and 1. The larger the difference in ensembles, the closer the value of $\eta$ is to 1. $\eta$ is normalized with respect to amino acid size, which allows comparison of $\eta$ between different amino acids. We found that for most amino acids $\eta$ < 0.5 (Fig 6C). If conformational ensemble shifts were purely in mean positions, and thermal fluctuations were of the order of 1.0 Å, then $\eta$ = 0.5 would correspond roughly to a shift in mean position by 0.7 Å. Therefore, amino acids with $\eta$ < 0.5 were considered to have small induced shifts. We found that amino acids with $\eta$ > 0.5 (the most affected with $\eta$ ~ 0.7) are close to RecA-D1 (Fig 6C). This suggests that the signal that originates at the P868 substitution site at the periphery of RecA-D2 [21] could allosterically reach the catalytic cleft and the linker between the two lobes, explaining the potential effect of P868L on RecA-D1 orientation.

## Discussion

BLM P868L and BLM G1120R both exist in the human population and are not associated with BS. However, both of these mutations cause increased SCEs and decreased recovery from hydroxyurea at low concentrations and decreased clearance of phosphorylated histone H2AX after treatment with camptothecin in human cells [22]. While these phenotypes are not as severe as what is observed in BS-causing mutations, these mutations still lead to genome instability and could cause an increased risk of cancer or other negative health outcomes. Many BS-causing BLM mutant proteins have been investigated *in vitro* and have been found to have decreased helicase activity and ATPase activity [17]. We hypothesized that these human BLM mutant proteins would lead to a moderate decrease in ATPase activity, DNA binding activity, and/or helicase activity.

Interestingly, BLM$_{core}$ P868L and BLM$_{core}$ G1120R are both active ATPases that have similar maximum ATPase rates to WT BLM$_{core}$ and require similar concentrations of ssDNA for ATPase activity stimulation (Fig 2, Table 2). Additionally, both mutant proteins have ATPase activity even in the absence of DNA, similar to WT BLM$_{core}$. When assessing the ability of these mutant proteins to bind to ssDNA, G4 DNA, and partial dsDNA, both were able to bind substrates with similar affinities to WT BLM BLM$_{core}$ While binding to ssDNA and G4 DNA was indicated by a smeared complex, binding to the partial duplex DNA induced a distinct band shift. The smearing observed with binding to ssDNA and G4 DNA substrates could indicate weaker overall binding and be caused by dissociation of BLM$_{core}$ from the DNA while running in the gel. The distinct band shift observed from the partial duplex is consistent with WT BLM$_{core}$, BLM$_{core}$ P868L, and BLM$_{core}$ G1120R binding more stably to the partial duplex than the other DNA constructs.

We also assessed the ability of BLM$_{core}$ P868L and BLM$_{core}$ G1120R to unwind dsDNA and G4-dsDNA substrates. WT BLM$_{core}$, BLM$_{core}$ P868L, and BLM$_{core}$ G1120R were able to unwind dsDNA efficiently and required similar amounts of BLM to reach half-maximum unwinding (Table 2). Interestingly, BLM$_{core}$ P868L was able to unwind G4-dsDNA more

efficiently than WT BLM$_{core}$, indicating that this mutant protein is slightly better at unwinding G4s than WT BLM$_{core}$. This could indicate that BLM$_{core}$ P868L has a tighter binding affinity for G4s than WT BLM$_{core}$, causing more proficient unwinding. However, direct binding studies did not demonstrate a higher affinity of BLM$_{core}$ P868L for G4s than WT BLM$_{core}$ (Table 2), so other properties such as processivity may be the source for this modest difference. BLM$_{core}$ G1120R is less efficient at unwinding the G4-dsDNA substrate than the dsDNA substrate alone, requiring more BLM$_{core}$ G1120R for half-maximum unwinding. This could be because this mutant protein is less efficient at unwinding G4s than dsDNA or because the BLM$_{core}$ G1120R mutant protein is less processive on longer substrates.

Overall, the *in vitro* activities of BLM$_{core}$ P868L and BLM$_{core}$ G1120R are strikingly similar to that of WT BLM$_{core}$, despite these mutant proteins causing moderate genome instability in cells. While it has been observed that BS-inducing BLM mutant proteins have decreased helicase and ATPase activity *in vitro*, this is not the case for these non-BS human BLM$_{core}$ mutant proteins [17]. However, there are several other ways that these mutations could cause genome instability in cells. Many RecQ helicases, including BLM, are regulated by posttranslational modifications (PTMs), such as phosphorylation, acetylation, and SUMOylation [3]. For example, BLM is phosphorylated in response to replication associated DNA damage or dsDNA breaks. Mutations that impact phosphorylation of BLM can decrease the ability of cells to recover from hydroxyurea exposure and other DNA damage [1]. Residues 868 and 1120 are not known sites of PTMs, making it unlikely that this is the reason that these mutant proteins cause increased SCEs [43]. However, residue K873, which is part of the lysine rich loop proximal to P868 is a reported site of SUMOylation and ubiquitination [43]. Because P868 terminates a β-strand at the start of the lysine-rich loop, it is possible that the mutation to leucine changes the structure of the loop and disrupts recognition of K873 by cellular PTM enzymes. This could result in decreased BLM activity in cells but more limited impact on *in vitro* activity of BLM.

Another possibility is that these mutations lead to increased SCEs in cells by producing BLM mutant proteins with disrupted dHJ dissolution activity and/or altered interaction with other proteins. BLM interacts with over 20 different proteins, including other dissolvasome proteins that are involved in dissolving dHJs [1, 3, 12]. If the BLM dissolvasome function is impaired due to weaker interactions between BLM and Top3α, RMI1, or RMI2 because of these mutations, this could lead to increased genome instability. From our studies, we cannot determine if P868L or G1120R human mutations decrease protein-protein interactions of BLM, but this will be an interesting target for future studies. From the MD simulations, a weakened interaction between DNA and the WH domain was observed for BLM P868L. This difference could contribute to the partial loss of cellular function of BLM P868L by disrupting BLM P868L dHJ resolution. Additionally, the change in orientation of the RecA-D1 lobe, which affects the catalytic cleft of the BLM helicase core, as well as subtle conformational changes in conserved helicase motifs V and VI (S3 Fig), could contribute to the functional deficits of BLM P868L in yeast and in human cells [21, 22]. However, the effects of these changes may be too small to be detected by the *in vitro* biochemical assays used in this study and would be more likely to have observable impact on other *in vivo* functions of BLM.

We also investigated the effects of substitutions to the lysine-rich loop proximal to the P868 residue in the BLM$_{core}$ K869A K870A construct. BLM$_{core}$ K869A K870A is moderately less stable than WT BLM$_{core}$, but still has a T$_m$ well above the temperatures used for our *in vitro* assays, indicating that this protein was stably folded in our studies (Fig 5). This BLM$_{core}$ K869A K870A mutant protein had a 4-fold increase in ssDNA concentration required for half-maximum stimulation of ATPase activity and a loss of ATPase activity in the absence of DNA. While DNA is known to stimulate ATPase activity of RecQ helicases, it was surprising that

BLM$_{core}$ K869A K870A had no ATPase activity in the absence of DNA. It is important for helicases to couple ATP hydrolysis to their functions of DNA translocation and DNA unwinding, and it has been found in the bacterial RecQ that the aromatic-rich loop (ARL) (residues 798–808 in BLM) is important for coupling ATP hydrolysis to the unwinding activity of RecQ. Substitutions to the RecQ ARL produce mutant proteins that either require more DNA for maximum ATP hydrolysis or with increased DNA-independent ATPase activity [44]. While the BLM$_{core}$ K869A K870A substitutions are not part of the ARL of BLM, this loop might have a similar effect on regulating ATPase activity and ATP-coupled functions of BLM. The MD simulations suggest that P868L has allosteric effects that alter the RecA-D1/D2 interface that forms the catalytic cleft of the BLM helicase domain. There could be multiple allosteric signaling pathways that connect the P868 site, and likely the lysine-rich loop, to the RecA-D1 lobe [45]. Identifying these signaling pathways and hotspots will require additional simulations [42], and expanding the simulations to other functionally impaired BLM mutants, including G1120R and mutants of the lysine-rich loop, will require additional simulations, and we expect this to be a target area for future studies.

In addition to the diminished ATPase activity, BLM$_{core}$ K869A K870A has a decreased apparent affinity for ssDNA and G4 substrates. Further, BLM$_{core}$ K869A K870A is deficient in stably and selectively binding to ssDNA-dsDNA junctions. While WT BLM$_{core}$ band shifts partial dsDNA, BLM$_{core}$ K869A K870A only causes a smear above the substrate. This smear is consistent with an overall reduced DNA binding stability for BLM$_{core}$ K869A K870A. Additionally, at high concentrations of BLM$_{core}$ K869A K870A, there is the appearance of a band at the size of the unannealed fluorescent-labeled ssDNA. This band appears to be the result of some amount of ATP-independent unwinding of the substrate from BLM$_{core}$ K869A K870A. If the lysine-rich loop of BLM is important for linking ATPase activity of BLM to its other functions, such as helicase activity, it is possible that substitutions to this loop are able to decouple these activities and generate a mutant protein that has weak ATP-independent helicase activity. Despite this, BLM$_{core}$ K869A K870A is unable to unwind the same amount of dsDNA or G4-dsDNA as WT BLM$_{core}$. BLM$_{core}$ K869A K870A does not reach over 80% unwinding of either substrate and requires higher concentrations of protein to unwind both dsDNA and G4-dsDNA. This may be due to enhanced strand annealing in the mutant protein, as has been observed with BLM previously [46]. BLM$_{core}$ K869A K870A requires approximately 2-fold more protein to unwind the G4-dsDNA compared to the dsDNA substrate. This could be because this mutant protein has reduced G4 binding or unwinding or that this mutant protein is less processive than WT BLM$_{core}$.

From this work, we conclude that BLM$_{core}$ P868L and BLM$_{core}$ G1120R are competent helicases in the *in vitro* activities tested herein. However, since both of these mutant proteins lead to genome instability phenotypes in human cells, these mutant proteins are clearly defective in some activities that are essential for genome stability. Future studies should focus on assessing the protein interaction network, dHJ dissolution activity, and cellular localization of these mutant proteins. We have also shown that the lysine-rich loop of BLM is an important regulator of both ATPase activity and helicase activity of BLM. This linker could play a role in coupling ATP hydrolysis with the unwinding action of BLM and can provide a rich area for future BLM studies.

## Supporting information

**S1 Fig. Purity and nuclease test of WT BLMcore and BLMcore variants.** (A). Assessment of the purity of WT BLM$_{core}$ and BLM$_{core}$ variants. Left to right: Blue Prestained Broad Range Protein Standard (New England Biolabs), WT BLM$_{core}$, BLM$_{core}$ P868L, BLM$_{core}$ G1120R, and

$BLM_{core}$ K869A K870A loaded onto a 4–15% Mini-PROTEAN TGX Precast Protein gel (BioRad). Each protein was at a concentration of ~2.5 μM. (B). Assessment of nuclease contamination of WT $BLM_{core}$ and $BLM_{core}$ variants at 1 μM concentration. Proteins were incubated with 40 nM fluorescein labeled $dT_{30}$ in 50 mM Tris-HCl, pH 7.5, 50 mM KCl, 1 mM DTT, 0.1 mg/mL Bovine Serum Albumin (BSA), and 5 mM $MgCl_2$ for 30 minutes at room temperature. Five μL of stop buffer (2% SDS, 5 μg/mL proteinase K, 20% (v/v) glycerol, 0.1 ethylenediaminetetraacetic acid (EDTA)) was added to each sample, and 5 μL of each sample was loaded onto a 15% acrylamide 1.5-mm gel in Tris-Borate-EDTA (TBE) buffer supplemented with 100 mM KCl. Gels were run at 75 V for 1 hour at 4˚C in 1xTBE running buffer with 100 mM KCl and imaged on an Azure c600 (Azure Biosystems). Samples in gel (from left to right) are DNA alone, WT BLM, BLM P868L, BLM G1120R, and BLM K869A K870A.
(TIF)

**S2 Fig. Normalized melting curves for each $BLM_{core}$ construct.** The normalized melt curves between 30˚C and 60˚C is shown for $BLM_{core}$, $BLM_{core}$ P868L, $BLM_{core}$ G1120R, and $BLM_{core}$ K869A K870A in the absence (blue circles) or presence of 0.5 mM ADP (red squares) or ATPγS (green triangle). Points on graph indicate the mean of three replicates and error bars indicate standard deviation calculated by Prism (version 9.3.1).
(TIF)

**S3 Fig. P868L-induced shift ($\eta$) in conformational ensemble of lobe 2 (D2 domain) of the ATPase domain of human BLM.** The X-ray structure of the D2 domain is shown as a cartoon putty where the width of the putty denotes the magnitude of $\eta$. The three conserved helicase motifs present in D2 and the linker connecting D2 to D1 are highlighted in color. The red sphere indicates the position of the P868 residue at the opposite side of D2.
(TIF)

## Acknowledgments

The authors thank members of the Keck laboratory for critical reading and evaluation of the manuscript.

## Author Contributions

**Conceptualization:** Rachel R. Cueny, Sameer Varma, Kristina H. Schmidt, James L. Keck.

**Formal analysis:** Rachel R. Cueny, Sameer Varma.

**Funding acquisition:** James L. Keck.

**Investigation:** Rachel R. Cueny, Sameer Varma, Kristina H. Schmidt, James L. Keck.

**Methodology:** Rachel R. Cueny.

**Resources:** James L. Keck.

**Supervision:** James L. Keck.

**Writing – original draft:** Rachel R. Cueny, Kristina H. Schmidt, James L. Keck.

**Writing – review & editing:** Rachel R. Cueny, Kristina H. Schmidt, James L. Keck.

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
