## [Decision Letter · Decision Letter 0]

21 Mar 2023

PONE-D-23-02128Biochemical Properties of Naturally Occurring Human Bloom Helicase VariantsPLOS ONE

Dear Dr. Keck,

Thank you for submitting your manuscript to PLOS ONE. After careful consideration, we feel that it has merit but does not fully meet PLOS ONE’s publication criteria as it currently stands. Therefore, we invite you to submit a revised version of the manuscript that addresses the points raised during the review process.

We look forward to receiving your revised manuscript.

Kind regards,

Hari S. Misra

Academic Editor

PLOS ONE

Reviewers' comments:

Reviewer's Responses to Questions

**Comments to the Author**

1. Is the manuscript technically sound, and do the data support the conclusions?

Reviewer #1: Partly

Reviewer #2: Yes

2. Has the statistical analysis been performed appropriately and rigorously? 

Reviewer #1: Yes

Reviewer #2: Yes

3. Have the authors made all data underlying the findings in their manuscript fully available?

Reviewer #1: Yes

Reviewer #2: Yes

4. Is the manuscript presented in an intelligible fashion and written in standard English?

Reviewer #1: Yes

Reviewer #2: Yes

5. Review Comments to the Author

Reviewer #1: Biochemical Properties of Naturally Occurring Human Bloom Helicase Variants

The authors present a set of biochemical experiments that set out to test the effects of two single nucleotide polymorphisms on the in vitro activities of recombinant human BLM helicase; including unwinding of different DNA substrates, EMSAs, ATPase assays and molecular dynamics calculations.

MAJOR POINTS

Page 6, Line 125 – ‘We also measured activity of a BLMcore K869A K870A mutant protein to assess the role of the lysines within the lysine-rich loop’.

>> Examination of the indicated PDB deposition, indicates that the sidechain of Lys870 is not involved in any polar interactions with the bound substrate (but provides a van der Waals surface), whereas Lys869 is involved in a complex series of internal stabilising hydrogen bonds, including the sidechain and backbone of Ser965, and the sidechain of Asp997. It is therefore not clear why mutations in these residues were chosen for examination in this study. Lys872 (and perhaps 873) due to the proximity to the phosphodiester backbone may have been more rational choices.

Page 8, Line 178 — ‘The BLMcore K869A K870A construct was purified with the following modifications to remove a contaminating nuclease’.

>> This causes some concern, suggesting that the introduced mutations have actually affected protein fold (esp. K869A, see comments above) leading to non-specific hydrophobic interactions with other cellular proteins. An assessment of the stability of the protein fold is warranted, given this data, i.e. a thermal denaturation / thermofluor / DSF type of experiment.

Page 11 — Molecular dynamic simulations

>> It is not fully clear how these simulations contribute to the overall conclusions of the paper. To a non-expert, the data presented in Figure 5 is generally not interpretable / easily understood. Also, to the non-trained eye the data presented in the lower section of panel A looks identical, with the overlay of the structure just serving to obfuscate the presented data. The authors also make several statements in the manuscript along the lines of “changes were not observed in our biochemical assays” when referring to predictions made by the simulation, again questioning its value / validitly.

Page 13 — ‘BLMcore K869A K870A was found to have a modestly different maximum ATPase rate compared to WT BLMcore (P value = 0.0348)’.

>>Why is this change ascribed more weight/significance that that between WT BLMcore and BLMcore P868L (P value = 0.0116)? Also, a 35% difference between the maximum ATPase rates of WT BLMcore and BLMcore P868L is not a “relatively small difference”. This section needs revisiting and conclusions reconsidered.

Page 14 — Table 2, ATPase Data Summary

>>It is not clear why BLMcore K869A K870A has no measurable basal rate of ATP hydrolysis. This again points to potential protein fold issues (see points raised above).

Page 18 — Lines 421 to 422 / Figure 4

>> The data presented in Figure 4c indicates that BLMcore K869A K870A is only able to achieve unwinding of ~65% of the supplied substrate (as judged by the plateau region) of the plotted data. This again indicating problems with fold, i.e. not all the supplied protein is active / less stable or more likely is not fully soluble at higher concentration.

Discussion

>> This part of the manuscript is unfortunately rather weak and is generally overly speculative in nature, and provides very little insight into the effects of either the P868L or G1220R polymorphisms on the in vivo functions of BLM helicase; e.g. “While BLM G1220R functions similarly to WT BLM in our assays, the G1220R substitution could cause a decrease in BLM binding affinity for HJs.” — so why not test this hypothesis?

MINOR POINTS

Page 5, Line 109 —

>> Do either the P868L or G1220R polymorphisms introduce a rare codon, or are positioned at a known splice site, intron/exon boundary? What are the predicted effects on fold of either amino acid change (as predicted by ‘Site Directed Mutator” similar web-based analysis server”).

Figure 1

>> provides insufficient molecular detail and should ideally include indications of known hydrogen-bond / polar / hydrophobic / stacking interactions. Side chain representations for the amino acids flanking Gly1220 should also be shown (GSKS motif).

Figure 4, panel C

>> Please plot all data on axes with the same ranges.

Standard Error of the Mean

>> The authors need to clarify if the error bars represent standard error of the mean (SEM) or in fact 1 standard deviation (1 SD). SEM requires the mean from more than one experiment for calculations; do each of the indicated ‘three replicates’ contain more than one measurement?

Reviewer #2: Manuscript of Cuenty et is entitled as “Biochemical Properties of Naturally Occurring Human Bloom Helicase Variants”. This study is focussed on to assess the biochemical characterization of two non-BS BLM mutations e.g., P868L and G1120R. Previously, others have shown that mutation of these two aa led to defective DSB repair and enhanced sister chromatic exchange (22). In this manuscript, authors have tried assess the ATPase, DNA binding and helicase activity of these two mutations. Beside, they have assessed the effect of lysine rich surrounding near P868L. Overall, the current investigation is advancing the biochemical characterization of different heterozygous mutations in BLM and may have some relevance to understand the defective functions of P868L and G1120R BLM in genome instability at cellular level. Manuscript is well written but lacks novelty in term of correlating specific biochemical properties with the defects at cellular level. I have following concerns, which may be addressed to enhance the quality of the manuscript

1. Line 108-111: “……..Interestingly, two substitutions, P868L and G1120R, led to a partial loss in function, demonstrated by increased sensitivity to hydroxyurea compared to wild-type (WT) BLM. These mutations also cause a 4-5 fold increase in SCEs and increased sensitivity to hydroxyurea in human cells (22).” This statement is not fully correct. As per the citation (22), mutations at P868L and G1120R show no or moderate HU sensitivity. Authors need to correct this.

2. “……..There are over 1400 BLM mutations that exist in the human population, with 62 encoding for pathogenic mutations, 118 mutations considered benign, and the rest of uncertain or conflicting human health relevance (20).” Among so many pathogenic and non-characterized mutations, how authors have chosen to study P868 and G1120 mutations. The rationale of choosing these specific non-BS BLM mutations is not very convincing and needs proper justifications in the introduction of manuscript. For example, what mutations are known in helicase and RQC domain and why these two mutations, among all, were chosen for the current study

3. Line 295-297: “While BLMcore 295 P868L had an apparent increase in maximum ATPase rate compared to WT BLM (P value = 0.0116), there is a relatively small difference (35%) between the WT BLMcore and BLMcore P868L maximum ATPase rates (Table 2).” 35% difference is significant and not small.

4. Authors have done MD simulations of BLMcore P868L, which show some interesting observation that P868L has no major impact on DNA-D2/ZN distance but it enhances DNA-WH distance, which I personally feel has some effect on ATPase activity of P868L.

5. Since these mutations have no major impact on DNA binding and helicase activity but shows significant effect on SCE and CPT induced replication DSB repair (22), it seems these residues may have some role with reference to different DNA specific structures. Authors should assess the DNA binding activity of P868 and G1120 for replication fork and Holliday junction substrates.

6. Why authors have not done MD simulation for other mutations (G1220R and K869A-K870A)?

6. PLOS authors have the option to publish the peer review history of their article (what does this mean?). If published, this will include your full peer review and any attached files.

Reviewer #1: No

Reviewer #2: No

---

## [Author Response · Author response to Decision Letter 0]

10 May 2023

Please see attached file, "Response to Reviewers"

---

## [Editor Report · Decision Letter 1]

23 May 2023

Biochemical Properties of Naturally Occurring Human Bloom Helicase Variants

PONE-D-23-02128R1

Dear Dr. Keck,

We’re pleased to inform you that your manuscript has been judged scientifically suitable for publication and will be formally accepted for publication once it meets all outstanding technical requirements.

Kind regards,

Hari S. Misra, Ph.D.

Academic Editor

PLOS ONE
---

## [Editor Report · Acceptance letter]

26 May 2023

PONE-D-23-02128R1 

Biochemical Properties of Naturally Occurring Human Bloom Helicase Variants 

Dear Dr. Keck:

I'm pleased to inform you that your manuscript has been deemed suitable for publication in PLOS ONE. Congratulations! Your manuscript is now with our production department. 

Kind regards, 

on behalf of

Professor Hari S. Misra 

Academic Editor

PLOS ONE